# Mako: Semi-supervised continual learning with minimal labeled data via data programming

## Abstract

Lifelong machine learning (LML) is a well-known paradigm mimicking the human learning process by utilizing experiences from previous tasks. Nevertheless, an issue that has been rarely addressed is the lack of labels at the individual task level. The state-of-the-art of LML largely addresses supervised learning, with a few semi-supervised continual learning exceptions which require training additional models, which in turn impose constraints on the LML methods themselves. Therefore, we propose Mako, a wrapper tool that mounts on top of supervised LML frameworks, leveraging data programming. Mako imposes no additional knowledge base overhead and enables continual semi-supervised learning with a limited amount of labeled data. This tool achieves similar performance, in terms of per-task accuracy and resistance to catastrophic forgetting, as compared to fully labeled data. We ran extensive experiments on LML task sequences created from standard image classification data sets including MNIST, CIFAR-10 and CIFAR-100, and the results show that after utilizing Mako to leverage unlabeled data, LML tools are able to achieve $97\%$ performance of supervised learning on fully labeled data in terms of accuracy and catastrophic forgetting prevention. Moreover, when compared to baseline semi-supervised LML tools such as CNNL, ORDisCo and DistillMatch, Mako significantly outperforms them, increasing accuracy by $0.25$ on certain benchmarks.

## 1 Introduction

Since 1995, researchers have been actively seeking machine learning (ML) frameworks that can better mimic the human learning process by retaining memories from past experiences (Thrun & Mitchell, 1995; Liu, 2017). Hence, in contrast to traditional isolated ML where knowledge is never accumulated, the concept of lifelong machine learning (LML) is defined as training ML models for continual learning over task sequences, with a knowledge base to store information that could be helpful in the future. Under this definition, LML can be seen as continual transfer learning (Pan & Yang, 2015) with memory or sequential multi-task learning (Caruana, 1997).

Nevertheless, at the level of individual tasks, the current state-of-the-art of LML still remains largely supervised. A challenge that has rarely been mentioned is that labels can still be expensive and rarely available for each task, such that supervised training can hardly provide acceptable performance. For LML, performance is not only measured at task level, but also for the entire lifetime, with attributes such as resistance to catastrophic forgetting and space-efficiency of knowledge base.

We take a more generic approach to semi-supervised lifelong learning and develop a technique that can wrap around any existing continual learning algorithm. Specifically, we propose to use *data programming* (Ratner et al., 2016b) to label new data under the supervision of a set of weak labeling functions trained from a limited number of labeled data points. Data programming has proven successful for isolated learning, and so this is the first work integrating it with lifelong learning.

Our approach, called Mako, sits atop an existing lifelong learner to turn it into a semi-supervised learner. For each task, Mako takes as input a small labeled dataset plus an unlabeled dataset with no restriction on size, generates labels for the unlabeled data and then updates the lifelong learner supervised by both sets. This approach builds upon the widely used Snuba algorithm (Varma & Ré,

2016), and we show that its theoretical guarantees still hold in the lifelong setting. Mako's procedure leverages automatic hyperparameter tuning, data programming and confidence calibration, in order to adapt automatic labeling to constantly varying LML tasks. We show that Mako produces adequately high performance, in terms of both task accuracy and prevention of catastrophic forgetting, that approaches that of training on fully labeled data without costing extra knowledge base space overhead.

We extensively evaluated Mako on various partially labeled LML task sequences generated from commonly used datasets including MNIST, CIFAR-10 and CIFAR-100 (LeCun & Cortes, 2010; Krizhevsky, 2009), while mounting on a diversity of supervised LML frameworks such as Deconvolutional Factorized CNN (DF-CNN), Dynamic Expandable Network (DEN) and Tensor Factorization (TF) (Lee et al., 2019; Yoon et al., 2018; Bulat et al., 2020). We compared the results to supervised learning on fully labeled data, as well as the same partially labeled data with existing semi-supervised LML tools: CNNL, ORDisCo and DistillMatch (Baucum et al., 2017; Wang et al., 2021; Smith et al., 2021). Empirically, we show that the performance of LML methods improves as giving more training data even with Mako labels, achieving at least $97\%$ performance relative to LML methods trained on ground-truth labels and being able to beat existing semi-supervised LML tools by approximately $0.25$ higher accuracy.

**Contributions Summary:**

1. Adapting automatic label generation by semi-supervised learning/data programming to LML in the scenario where labeled training data is expensive to obtain compared to unlabeled for all tasks.

2. Implementing a LML wrapper, namely Mako, for the scenario of expensive labels, such that, when given partially labeled data of limited size and compared to the current supervised state-of-the-art with fully labeled data, each LML task sequence (1) requires significantly fewer labeled data to achieve high accuracy on individual tasks, (2) maintains similar resistance to catastrophic forgetting and (3) does not cost extra knowledge base storage.

3. Extensive experiments on lifelong binary and multiclass image classification on 3 commonly used datasets, including comparison to supervised and semi-supervised LML baselines, accomplishing very close performance to supervised LML on fully labeled data and significantly outperforming baseline semi-supervised LML frameworks.

## 2 RELATED WORK

Lifelong machine learning (LML) is a concept of continual learning among a sequence of tasks. Specifically, given multiple machine learning tasks arriving continuously, the LML framework accumulates the knowledge of previous tasks and utilizes it for future ones (Chen & Liu, 2016; Ruvolo & Eaton, 2013b). The method of retaining and adopting the knowledge base varies depending on the characteristics of tasks and learning models. For instance, a latent matrix is used to keep prototypical weights for linear models (Kumar & Daume, 2012; Ruvolo & Eaton, 2013a), and for deep neural networks, either weights of layers (Yoon et al., 2018; Lu et al., 2017; Bulat et al., 2020; Lee et al., 2019) or extracted features of layers (Rusu et al., 2016; Misra et al., 2016; Gao et al., 2019; Schwarz et al., 2018) are considered as the knowledge base to transfer. Besides research on models to transfer knowledge across tasks, the LML community has also been working on task order sorting for higher overall performance (Sun et al., 2018; Ruvolo & Eaton, 2013b). Nevertheless, all these works assume that each individual task training is supervised with ground-truth labels for the entire training set. In many real-life ML tasks, obtaining labels has been and continues to be expensive (Settles, 2009). Only a few works have addressed this issue on LML in very specific domains, such as continual sentimental analysis with words partially labeled (Wang et al., 2016) and reinforcement learning in a sequence of state spaces without explicit rewards (Finn et al., 2017). Different from previous work, Mako enables high performance in continual learning on generic ML tasks given partially labeled data, with the cardinality of labels restricted at a small number.

To tackle expensive labels in ML, researchers have purposed various methods such as active learning (Settles, 2009) or semi-supervised learning (Olivier et al., 2006), which is an ML framework that takes both labeled and unlabeled data as inputs. The unlabeled data can directly assist the training, or can serve as a target data set yet to be labeled. In recent years, with the rising enthusiasm and need in deep learning, various methods have been used to assist semi-supervised learning with DNN models (Rasmus et al., 2015; Laine & Aila, 2016; Tarvainen & Valpola, 2017). One branch of

semi-supervised learning, namely data programming or weak supervision (Ratner et al., 2016b), focuses on auto-generation of labels on an unlabeled dataset with the help of a small labeled dataset. Targeting this problem, researchers leverage weak labeling functions that each can label with an accuracy slightly higher than a coin flip. Then, ensembling algorithms are able to bag these weak labelers and produce highly accurate final labels. In other words, the weak labels supervise the training of the final ensembled generative model. There exists different implementations of such generative model training such as Snorkel (Ratner et al., 2016a).

Semi-supervised learning is an underexplored problem in the LML setting; all the works mentioned above have addressed problems in isolated ML only. Early work in semi-supervised incremental learning, in which all tasks are trained simultaneously as batches of unlabeled data are incrementally observed by a classifier, has identified challenges pertaining to selection of the label generating model architecture and confidence calibration of the generated labels (Baucum et al., 2017), processes which Mako undertakes autonomously through automatic search for weak labelers before ensembling, and confidence calibration after labels are produced. Pioneering work in semi-supervised LML requires training of additional models for out-of-distribution detection (Smith et al., 2021) or generative replay (Wang et al., 2021). Conversely, by focusing on label generation through data programming, our approach achieves similar behavior through a pre-processing workflow that does not impose additional constraints on the LML approach or require additional knowledge base storage.

For data programming tools, it has been a long standing issue to obtain weak labeling functions as input. One approach is manually designing the weak labelers by domain experts (Ratner et al., 2016b;a), while other researchers seek automatic labeler generation. For instance, Snuba (Varma & Ré, 2016) outputs weak supervisors with an accuracy guarantee on the final ensembled data programming model. Thus far, Snuba has been evaluated on various isolated ML tasks, with weak labelers in form of k-nearest-neighbors, decision stumps and logistic regressors, and has been shown to ace in natural language processing and image classification. It also succeeds in more complex applications such as classifying bone tumors in medical domain. In our work, we build a weak labeling function generation module in Mako based on Snuba, aiming to support a larger variety of labeler models for LML, while maintaining the theoretical accuracy guarantee of generated labels.

## 3 PROBLEM FORMULATION

**LML:** Given a sequence of ML tasks: $T_1, T_2, \ldots, T_m$, where $m$ is an unbounded integer, each task $T_i$ has an underlying data space $\mathcal{X}_i$ and label space $\mathcal{Y}_i$. $T_i$ has $n_{i,train}$ labeled training data $(X_{i,train}, Y_{i,train}) \sim \mathcal{D}_i$, such that $X_{i,train} \in \mathcal{X}_i^{n_{i,train}}$ and $Y_{i,train} \in \mathcal{Y}_i^{n_{i,train}}$, following some task-specific i.i.d. distribution $\mathcal{D}_i$. The task also has $n_{i,test}$ testing data $(X_{i,test}, Y_{i,test})$ drawn from the same distribution.

Traditionally, an LML model encounters the tasks in order. At $T_i$, it trains on the full set of $(X_{i,train}, Y_{i,train})$ with supervised learning and obtain model $f_i : \bigcup_{j=1}^{i} \mathcal{X}_i \mapsto \bigcup_{j=1}^{i} \mathcal{Y}_i$. Immediately after this training, it predicts the $j$-th task with $\hat{Y}_{ij} = f_i(X_{j,test})$ for each $j \leq i$. There are two objectives:

1. High test accuracy. Formally, denote testing accuracy of task $T_j$ right after the training of task $T_i$ on $(X_{i,train}, Y_{i,train})$ as

$$a_{ij} = \frac{\sum_{k=1}^{n_{j,test}} \mathbf{1}(\hat{y}_k = y_k)}{n_{j,test}} \tag{1}$$

with $\hat{y}_k$ and $y_k$ the $k$-th label in $\hat{Y}_{ij}$ and $Y_{i,test}$, respectively. One LML objective in the current state-of-the-art is to maximize the **peak per-task accuracy up to task i** defined as

$$\tilde{a}_i = \frac{1}{i} \sum_{j=1}^{i} a_{jj} \tag{2}$$

Peak per task accuracy is defined on all tasks $i = 1, \ldots, m$, and shows the LML performance on the current training task on the basis of the accumulated knowledge.

2. Low catastrophic forgetting. A critical aspect of LML is to prevent catastrophic forgetting of previous tasks. The resistance to catastrophic forgetting can be quantified by the average testing accuracy of all tasks encountered so far. For each task $T_i$, denote a second objective, **final accuracy**

**up to task i** as

$$\bar{a}_i = \frac{1}{i} \sum_{j=1}^{i} a_{ij} \tag{3}$$

The second objective is to maximize $\bar{a}_i$ for all $i = 1, \ldots, m$. The gap between peak per-task accuracy and final accuracy expresses how much knowledge of previous tasks LML method can preserve through the streams of tasks. Therefore, we define a third metric to be maximized, **catastrophic forgetting ratio up to task i**, as

$$c_i = \frac{1}{i} \sum_{j=1}^{i} \frac{a_{ij}}{a_{jj}} \tag{4}$$

This metric is less than 1 if the LML model loses its performance on the earlier tasks, and it is greater than 1 if there is positive knowledge transfer from the later tasks to the earlier ones.

**Semi-supervised LML with data programming:** The key challenge we want to address is the cost of labeling data in all LML tasks. That is, the number of available labeled training data is small. Therefore, we would like to split $(X_{i,train}, Y_{i,train})$ in the traditional LML problem into disjoint $(X_{i,L}, Y_{i,L}, X_{i,U})$, where $(X_{i,L}, Y_{i,L})$ is a set of labeled training data with restriction on size and $X_{i,U}$ is a set of unlabeled data with no such restriction. In other words, $|X_{i,L}|$ is small, e.g. $|X_{i,L}| \leq 150$ for a 10-way image classification problem.

Apparently, the LML model is now trained with semi-supervised learning. For task $T_i$, we would like to leverage data programming on $(X_{i,L}, Y_{i,L}, X_{i,U})$ to first train a generative model $\pi_i : \mathcal{X}_i \mapsto \mathcal{Y}_i$. This model automatically labels $X_{i,U}$ with $Y_{i,U}^\pi = \pi_i(X_{i,U})$. Then, the LML framework will be trained on $(X_{i,L}, Y_{i,L}, X_{i,U}, Y_{i,U}^\pi)$. Denote the LML model immediately after this training as $f_i^\pi : \bigcup_{j=1}^{i} \mathcal{X}_i \mapsto \bigcup_{j=1}^{i} \mathcal{Y}_i$, and $\hat{Y}_{ij}^\pi = f_i^\pi(X_{j,test})$ for each task $j \leq i$.

In this scenario, we redefine testing accuracy in Equation 1 as

$$a_{ij}^\pi = \frac{\sum_{k=1}^{n_{j,test}} \mathbf{1}(\hat{y}_k^\pi = y_k)}{n_{j,test}} \tag{5}$$

where the only difference is that $\hat{y}_k^\pi$ is the $k$-th label in $\hat{Y}_{ij}^\pi$ instead of $\hat{Y}_{ij}$. From Equation 5, we can then redefine metrics in Equations 2, 3 and 4 as

$$\tilde{a}_i^\pi = \frac{1}{i} \sum_{j=1}^{i} a_{jj}^\pi \quad (6) \qquad \bar{a}_i^\pi = \frac{1}{i} \sum_{j=1}^{i} a_{ij}^\pi \quad (7) \qquad c_i^\pi = \frac{1}{i} \sum_{j=1}^{i} \frac{a_{ij}^\pi}{a_{jj}^\pi} \quad (8)$$

Our research problem is to design a semi-supervised LML framework leveraging data programming that minimizes the performance between using partially labeled data, $(X_{i,L}, Y_{i,L}, X_{i,U}, Y_{i,U}^\pi)$, and the upper-bound performance using fully labeled data, $(X_{i,L}, Y_{i,L}, X_{i,U}, Y_{i,U})$, in terms of these three metrics while imposing no additional knowledge base storage overhead in the LML framework.

## 4 MAKO: LIFELONG MACHINE LEARNING FRAMEWORK

We implemented Mako, a semi-supervised LML framework that can be mounted on top of any existing LML tool, with $(X_{i,L}, Y_{i,L}, X_{i,U})$ as input for each task $T_i$. The automatic label generation of $Y_{i,U}^\pi$ is done by data programming, consisting of generating weak labeling functions and ensembling a generative model. Nevertheless, different from data programming on isolated ML, due to the variation of tasks throughout LML, we need to actively adapt both the weak and the strong models to maximize the performance at each task. This is accomplished by automatic search on architecture and training hyperparameter search before weak labeler generation and confidence calibration after the ensembled model $\pi_i$ is trained. The following subsections explains the Mako workflow step-by-step, as illustrated in Figure 1.

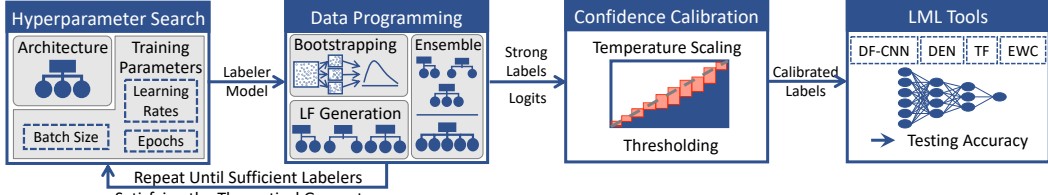

Figure 1: Mako Framework

## 4.1 AUTOMATIC HYPERPARAMETER SEARCH

Before producing weak labeling functions, two questions must be answered: (1) How do we choose the model architecture for our weak labeling functions? (2) How do we train these models after architecture is decided? Formally, these questions lead to the decision of architectural and training hyperparameters of labelers, which Mako accomplishes automatically.

The only prior knowledge required is a search space, manually designed, of the architecture and training hyperparameters. For instance, given that the task is image classification, the search space of labeler architecture will include CNNs with the same input and output dimensions, but different configurations of hidden layers. Similarly, the training hyperparameters will be searched on different learning rates, batch sizes and number of epochs.

The automatic tuning of weak labelers therefore traverses the search space until a satisfactory configuration is found. The stopping criterion is defined as follows: after training ten labelers with a given configuration of architecture and training hyperparameters on different bootstrapped subsets of $(X_{i,L}, Y_{i,L})$, 9 out of 10 achieves an accuracy on the entire $(X_{i,L}, Y_{i,L})$ above some threshold $a$. This criterion shows the generalization capability of sampled labelers and alternative criteria can be used, such as 8 out of 10 or average accuracy $\geq a$. We accept such a configuration because we train the labelers on bootstrapped subsets of data during data programming, which will be further explained in 4.2. If no configuration in the search space is satisfactory, we decrease $a$ by 0.05 and repeat the search. Hence, we can start from a sufficiently high $a$ such as 0.9. For efficiency, Mako selects the configuration with the smallest number of training epochs among all the acceptable ones.

There exists a efficiency-automation trade-off in this hyperparameter search. The less prior knowledge we have for the task, the larger search space we need to input to Mako and thus a larger time overhead. In contrast, more manual exploration on the task will lead to a smaller search space and therefore efficiency.

## 4.2 GUARANTEED DATA PROGRAMMING WITH BOOTSTRAPPING ON TRAINING SET

After deciding the architecture and training hyperparameters of weak labeling functions, Mako kicks off weak labeler training and ensembling of generative model $\pi_i$ for each task $T_i$. For the training process, we are inspired by Snuba (Varma & Ré, 2016) due to the fact that it has a theoretical accuracy guarantee in the final generative model. Nonetheless, in order to increase variance among the labelers, Snuba trains them on different selected subsets of features. This procedure does little favor in LML, where we might encounter complex tasks in the sequence that requires the entire feature space. For instance, in image classification, excluding pixels in training may lead to a failure in capturing important feature-target relationships, especially when we want to take advantage of local connectivity of CNNs. Therefore, Mako uses bootstrapping on $(X_{i,L}, Y_{i,L})$ and trains each labeler with a sampled subset. The bootstrapping size can be either manually given or searched as a training hyperparameter.

Besides bootstrapping on data, the rest of weak labeler generation process remains unchanged from Snuba in order to maintain its theoretical guarantee. Snuba has such a guarantee because it checks an exit condition on whether the committed labelers have labeled sufficient number of data points in $X_{i,L}$ with acceptable confidence. If so, these labelers guarantee that the final generative model will have a learned accuracy on $X_{i,L}$ close to the labeling accuracy on $X_{i,U}$. Formally, this guarantee can be stated as Proposition 1.

**Proposition 1 (Theoretical Guarantee of Snuba):** *Given $h$ committed weak labelers, denote the empirical accuracies of all committed weak labelers on $X_{i,L}$ as a vector $a_{i,L,w}$. The generative*

*model training outputs learned accuracies of these labelers on $X_{i,L}$ as $a_{i,L,g}$ before ensembling. Moreover, the unknown true accuracies of the labelers on $X_{i,U}$ after generative model training are denoted as $a_{i,U,g}$, with $a_{i,L,w}, a_{i,L,g}, a_{i,U,g} \in \mathbb{R}^h$. We have a measured error $||a_{i,L,w} - a_{i,L,g}||_\infty \leq \epsilon$. Then, if each labeler labels a minimum of*

$$d \geq \frac{1}{2(\gamma - \epsilon)^2} \log \left( \frac{2h^2}{\delta} \right) \tag{9}$$

*data points in $X_{i,L}$ with above some given confidence threshold $\nu$ for all iterations, we can guarantee that $||a_{i,U,g} - a_{i,L,g}||_\infty < \gamma$ for all iterations with probability $1 - \delta$.*

The proof of Proposition 1 from the original Snuba paper is included in Appendix A. We would like to show that our bootstrapping technique maintains this guarantee, as in Proposition 2.

**Proposition 2 (Theoretical Guarantee of Mako Weak Labeler Generation):** *Despite the modifications from Snuba, the Mako weak labeler generation process satisfies the theoretical guarantees described in Proposition 1.*

**Proof of Proposition 2:** Although our weak labelers are generated with (1) different model architectures and (2) sampling by bootstrapping on data instead of features, it still satisfies the definition of weak heuristics required by Snuba. That is, a function $f : \mathcal{X}_i \mapsto \mathcal{Y}_i \cup \{abstain\}$, where $abstain$ is a special label assigned to data points with a confidence lower than the threshold $\nu$.

Consequently, by Proposition 2, if Mako checks the same theoretical exit condition at each iteration, the final generative model satisfies the same theoretical guarantee as in Snuba. The hyperparameter search described in 4.1 can be understood as looking for labelers with high $a_{i,L,w}$. Based on this guarantee, we will have $||a_{i,U,g} - a_{i,L,w}||_\infty \leq \epsilon + \gamma$ with probability $1 - \delta$, and hence we can probablistically achieve high labeling accuracy $a_{i,U,g}$.

The weak labeling functions are then used to label both $X_{i,L}$ and $X_{i,U}$ and the labels are input for the training of generative model $\pi_i$. We label $X_{i,L}$ as well for confidence calibration, which will be explained in the following subsection.

### 4.3 CONFIDENCE CALIBRATION AND THRESHOLDING

Considering the variability and complexity of LML tasks, we believe the generative model labels can be further improved to approach the ground truth. Therefore, Mako takes an extra step to adjust the trained generative model $\pi_i$ by confidence calibration on the produced logits on $X_{i,U}$.

Confidence calibration (Guo et al., 2017) is the process to align the confidence of labeling a data point to the accuracy, which is an estimation of the probability that the data point is labeled correctly. The procedure of confidence calibration is adjusting the logits in order to minimize the expected calibration error (ECE), formalized as

$$\min_{p_{k,i,U}^\pi} \mathrm{E}_{p_{k,i,U}^\pi}[|\Pr[y_{k,i,U}^\pi = y_{k,i,U} \mid p_{k,i,U}^\pi = p] - p|] \tag{10}$$

where $y_{k,i,U}$ is the ground truth label of the $k$-th data point in $X_{i,U}$, $y_{k,i,U}^\pi$ is its predicted label by the data programming generative model and $p_{k,i,U}^\pi$ is the associated confidence. However, since the ground truth labels of $X_{i,U}$ is unknown, this approach is infeasible. This is the reason for training the generative model under the supervision of weak labels on both $X_{i,U}$ and $X_{i,L}$. The model is hence expected to have the similar ECE on labeled and unlabeled data. Mako then performs temperature scaling (Platt, 1999) to minimize ECE on $X_{i,L}$, and then apply the same temperature on $X_{i,U}$ to adjust the logits.

After calibration, there could still exist data points labeled with low confidence. We then threshold $X_{i,U}$ with a confidence of $1/n_{classes} + \beta$ for some small positive $\beta$. For example, in a 10-way classification problem, pick $\beta = 0.01$. We only keep the subset of $X_{i,U}$ with calibrated confidence $\geq 0.1 + 0.01 = 0.11$. Denote this kept portion of unlabeled data and labels as $(X_{i,U,c}, Y_{i,U,c}^\pi)$.

### 4.4 LML TASK TRAINING AND EVALUATION

Finally, $X_{i,L}$, $Y_{i,L}$, $X_{i,U,c}$ and $Y_{i,U,c}^\pi$ will be input into the mounted LML tool for the training of task $T_i$. The evaluation of the current model will be performed on all the hold-out testing data set $(X_{test,1}, Y_{test,1}), \ldots, (X_{test,i}, Y_{test,i})$.

Mako does not alter the internal procedures of the LML tool. Instead, it generates labels for unlabeled data and then feeds it to the LML tool as a black box. This design enables high modularity and the LML tool can be easily replaced. In the experiment section, we will demonstrate Mako's performance when mounting on various of LML paradigms such as DF-CNN, DEN and TF. This design also imposes no additional knowledge space storage in LML.

## 5 EXPERIMENTS

### 5.1 DATASETS AND LML TASK SEQUENCES

We evaluate Mako on LML task sequences generated from 3 commonly-used image classification datasets: MNIST (LeCun & Cortes, 2010), CIFAR-10 and CIFAR-100 (Krizhevsky, 2009).

We mount Mako on existing supervised LML tools: DF-CNN, and TF (Lee et al., 2019; Bulat et al., 2020) to enable semi-supervised learning, and compare the performance to supervised LML on labeled data only as well as all data fully labeled. The task sequences are:

1. Binary MNIST: for each task, pick classes 0 vs 1, 0 vs 2, ..., 8 vs 9, with 45 tasks in total.

2. Binary CIFAR-10: same construction as binary MNIST.

Additionally, we compare Mako-mounted supervised LML tools to existing semi-supervised LML: CNNL, ORDisCo, and DistillMatch (Baucum et al., 2017; Wang et al., 2021; Smith et al., 2021). We generate the same task sequences as in their papers:

3. 10-way CIFAR-10: the entire CIFAR-10 as one task.

4. 5-way CIFAR-100: for each task, pick each superclass in CIFAR-100, with 20 tasks in total.

5. 5-way MNIST: for each task, pick classes 0-4 and 5-9 in MNIST, with 2 tasks in total.

Details of the tasks such as data splits, hyperparameters searched and labeling accuracy are explained in Appendix B.

### 5.2 COMPARISON TO SUPERVISED LML

For each task, data is split into a labeled training set, an unlabeled training set and a hold-out test set. The details of data split and weak labeler models searched are included in Appendix B.

Table 1 and 2 summarize the performance of LML methods according to the amount of training data: **L** standing for the labeled training set and **U** with number denoting the number of instances in the unlabeled training set used during training of each task. These tables show peak per-task accuracy, final accuracy and catastrophic forgetting ratio up to the last task (Eqn. 6, 7 and 8, respectively). Additionally, we trained the LML methods on the same data with true labels instead of the Mako-generated labels to quantify the quality of the generated labels. LML methods achieve better peak

| LML | Data | Performance | | | Relative to True Labels | |
|---|---|---|---|---|---|---|
| | | Per-Task Acc. | Final Acc. | Forget. Ratio | Per-Task Acc. | Final Acc. |
| TF | L | $95.6 \pm 0.4$ | $96.8 \pm 0.6$ | $1.02 \pm 0.01$ | - | - |
| | L+U30 | $95.9 \pm 0.4$ | $97.0 \pm 0.4$ | $1.01 \pm 0.01$ | 0.99 | 1.00 |
| | L+U60 | $96.1 \pm 0.3$ | $96.8 \pm 0.5$ | $1.01 \pm 0.01$ | 0.99 | 0.99 |
| | L+U120 | $96.2 \pm 0.2$ | $96.9 \pm 0.2$ | $1.01 \pm 0.00$ | 0.99 | 0.99 |
| | L+U240 | $96.5 \pm 0.2$ | $96.4 \pm 0.2$ | $1.00 \pm 0.00$ | 0.98 | 0.98 |
| DF-CNN | L | $93.8 \pm 0.4$ | $95.4 \pm 0.3$ | $1.02 \pm 0.01$ | - | - |
| | L+U30 | $94.6 \pm 0.3$ | $96.2 \pm 0.4$ | $1.02 \pm 0.01$ | 0.98 | 1.00 |
| | L+U60 | $94.9 \pm 0.2$ | $96.1 \pm 0.7$ | $1.01 \pm 0.01$ | 0.98 | 0.99 |
| | L+U120 | $95.2 \pm 0.3$ | $95.8 \pm 0.9$ | $1.01 \pm 0.01$ | 0.98 | 0.99 |
| | L+U240 | $95.9 \pm 0.2$ | $94.5 \pm 1.3$ | $0.99 \pm 0.01$ | 0.98 | 0.97 |

Table 1: Supervised LML experiments on binary MNIST tasks, showing mean $\pm$ standard deviation. LML models are trained on either **L**abeled data only or **L**abeled data and instances of **U**nlabeled data with Mako-generated labels, and evaluated three metrics up to task $45$ (Eqn. 6, 7 and 8) as well as accuracy metrics relative to LML models trained on the same data with true labels instead.

| LML | Data | Performance | | | Relative to True Labels | |
|---|---|---|---|---|---|---|
| | | Per-Task Acc. | Final Acc. | Forget. Ratio | Per-Task Acc. | Final Acc. |
| TF | L | $81.5 \pm 0.2$ | $76.8 \pm 0.7$ | $0.95 \pm 0.01$ | - | - |
| | L+U200 | $83.4 \pm 0.4$ | $76.8 \pm 1.6$ | $0.93 \pm 0.02$ | 0.99 | 1.00 |
| | L+U400 | $84.3 \pm 0.3$ | $76.7 \pm 1.4$ | $0.92 \pm 0.02$ | - | - |
| DF-CNN | L | $80.7 \pm 0.3$ | $80.4 \pm 0.7$ | $1.01 \pm 0.01$ | - | - |
| | L+U200 | $82.7 \pm 0.3$ | $80.0 \pm 0.5$ | $0.98 \pm 0.01$ | 0.99 | 0.99 |
| | L+U400 | $84.0 \pm 0.3$ | $80.0 \pm 0.4$ | $0.96 \pm 0.01$ | - | - |

Table 2: Supervised LML experiments on binary CIFAR-10 tasks, mean $\pm$ standard deviation.

per-task accuracy as more Mako-labeled data is provided for training. This shows that Mako is able to generate useful labels for training while the current task, and especially data distribution, keeps changing in the lifelong learning setting. Training with Mako labels has no significant improvement in final accuracy, but, compared to LML models trained on the true data, both peak per-task accuracy and final accuracy are at least $97\%$ of the counterparts. Learning curves for each experiment are shown in Appendix C.

## 5.3 Comparison to Semi-Supervised LML

As discussed in Section 2, there is little prior work in semi-supervised LML settings. We identify three approaches for comparison: CNNL, ORDisCo, and DistillMatch (Baucum et al., 2017; Wang et al., 2021; Smith et al., 2021). Each baseline approach contains multiple modules and networks which each require their own tuning; to enable fair comparisons, we replicate the experimental conditions (data set, amount of labeled vs. unlabeled data, task definitions, and network architecture) and compare our approach to the best results reported in each baseline's original publication. Specifically, we replicate the instance-incremental learning experiments of CNNL on MNIST and CIFAR-10, and the class-incremental learning experiments of ORDisCo and DistillMatch on CIFAR-10 and CIFAR-100, respectively. We briefly describe each experiment below; for more details, see Appendix B for data splits and the original publications for full experimental descriptions.

**Instance-Incremental Experiments.** We compare semi-supervised learning using Mako in an instance-incremental setting (i.e., where all tasks are present at every epoch, but subsequent epochs contain different batches of unlabeled data), using an identical two-convolutional-layer CNN as described in (Baucum et al., 2017). The MNIST experiment uses a labeled data set of 150 images, with 1000 unlabeled images introduced in each of 30 instance-incremental batches. The CIFAR-10 experiment uses a labeled data set of 2000 samples, with subsequent batches of 1000 unlabeled images. Each Mako experiment was run over 10 random seeds. For additional context, we include the performance of the same network trained in the instance-incremental setting where all data is labeled with the ground truth, representing an upper bound on semi-supervised performance. As shown in Table 3, using Mako labels in place of CNNL's semi-supervised labeler results in comparable classification accuracy with better sample efficiency (MNIST), or strictly higher classification accuracy (CIFAR-10). Learning curves for each experiment are shown in Appendix D.

| | MNIST | | CIFAR-10 | |
|---|---|---|---|---|
| Approach | Final Acc. | Batches to Saturation | Final Acc. | Batches to Saturation |
| CNNL | **90.0** | 26 | 45.7 | 25 |
| Mako Labeled | $\mathbf{90.0 \pm 0.4}$ | $3.3 \pm 1.6$ | $\mathbf{54.2 \pm 0.5}$ | $26.5 \pm 1.1$ |
| *Fully Labeled* | *$99.0 \pm 0.1$* | *$17.0 \pm 4.0$* | *$57.6 \pm 0.5$* | *$27.1 \pm 1.7$* |

Table 3: Instance-incremental semi-supervised LML experiments, showing mean $\pm$ standard deviation (where available). Final accuracy (see Eqn. 7) is measured on the data set's standard held out test set, batches to saturation is measured as the first epoch where a 3-batch sliding window average meets or exceeds the final accuracy

**Class-Incremental Experiments.** We compare state-of-the-art LML methods (DF-CNN, DEN, and TF (Lee et al., 2019; Yoon et al., 2018; Bulat et al., 2020)) using Mako labels to perform semi-supervised learning in a class-incremental setting (i.e., the model is sequentially presented with tasks containing new sets of classes). For all experiments below, each Mako-enabled approach was run

over 10 random seeds, and we also include the performance of the same using all ground truth labels. We first compare Mako to ORDisCo over five binary CIFAR-10 tasks using 400 labeled instances, with the remaining data unlabeled. We note that we could not directly replicate the ORDisCo classification network architecture due to a lack of details in the original publication, but instead show that using Mako labels achieve a higher total classification accuracy with DF-CNN and TF using a significantly smaller network (4 convolutional layers for the Mako approaches, compared to OR-DisCo's 9 convolutional layers), shown in Table 4. Additionally, we compare Mako to DistillMatch over the 5-way CIFAR-100 using 20% labeled data, replicating DistillMatch's ParentClass task. As shown in Table 4, all of the LML methods enabled by Mako labeling meet or exceed DistillMatch's performance over the learning task. Learning curves for each experiment are shown in Appendix D.

| | CIFAR-10 | | CIFAR-100 | |
|---|---|---|---|---|
| | Approach | Final Acc. | Approach | Final Acc. |
| Baseline | ORDisCo | 74.8 | DistillMatch | $24.4 \pm 0.4$ |
| Semi-Supervised | Mako-labeled DF-CNN | $\mathbf{86.8 \pm 1.3}$ | Mako-labeled DF-CNN | $\mathbf{50.0 \pm 0.8}$ |
| | Mako-labeled DEN | $61.4 \pm 2.3$ | Mako-labeled DEN | $24.1 \pm 0.6$ |
| | Mako-labeled TF | $82.1 \pm 2.7$ | Mako-labeled TF | $48.9 \pm 2.1$ |
| Supervised | *Fully-labeled DF-CNN* | *$86.8 \pm 1.4$* | *Fully-labeled DF-CNN* | *$52.4 \pm 1.4$* |
| | *Fully-labeled DEN* | *$61.4 \pm 3.7$* | *Fully-labeled DEN* | *$23.7 \pm 0.4$* |
| | *Fully-labeled TF* | *$82.5 \pm 4.6$* | *Fully-labeled TF* | *$51.0 \pm 1.8$* |

Table 4: Class-incremental semi-supervised LML experiments, showing mean $\pm$ standard deviation (where available). Final accuracy (see Eqn. 7) is measured on the data set's standard held out test set, evaluated on all tasks after the final task completes training. For a breakdown of individual task accuracy and catastrophic forgetting ratios, see Appendix D.

## 6 Conclusion and Future Work

In this paper, we identified the challenge that collecting task-level labeled data for LML is expensive. We address this challenge with data programming, where labels are automatically generated, and implemented the Mako framework that can be mounted on top of existing LML tools. Mako takes in a limited number of labeled data and an unlimited number of unlabeled data, aiming to minimize the performance gap to using the same data but fully labeled. We demonstrated Mako can achieve sufficiently high accuracy per task as well as resistance to catastrophic forgetting, while costing no additional knowledge base storage, over a set of common image classification LML task sequences.

Future work on this topic can consider how to better resolve the issue of expensive labels at individual LML tasks. This includes how to extend Mako to a larger variety of LML task sequences. For instance, what alternative methods can be used for labeler hyperparameter tuning other than searching, what prior knowledge of tasks could assist automatic labeling, and how should the efficiency-capability trade-off of different models be balanced. Outside of the Mako framework, alternative solutions can modify existing LML tools directly, compromising a certain degree of modularity while possibly improving the overall LML performance.

Another interesting future direction is to consider the scenario where the labeled data and unlabeled data input per task are drawn from different distributions. This can be enabled by optimal transport theory, specifically the minimization of Sinkhorn distance between two distributions (Cuturi, 2013). Ideally, this approach could allow unlabeled data from other datasets to assist LML, resolving the issue of not only expensive labels but also expensive data collection.

## 7 REPRODUCIBILITY STATEMENT

We encourage researchers to replicate our experiments. As discussed in Section 4, we have provided the step-by-step explanation of Mako workflow. We have included more experiment details in Appendix B for convenience. Additionally, we pushed a sample code to an anonymous GitHub repository (`https://github.com/mako-anon/mako`). All readers are welcomed to pull our code and execute Mako on the example data themselves.

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

## APPENDIX A  SNUBA THEORETICAL GUARANTEE AND PROOF

Proposition 1 in Section 4.2 and its proof are given by the original Snuba paper (Varma & Ré, 2016). We adapt their notations to our paper to support the propositions in Section 4.2 on an arbitrary LML task $T_i$. For convenience, we remove the subscript $i$ in all notations.

**Proof of Proposition 1:**   First, the probability of the $l_\infty$ error between the labeling accuracies of the generative model on $X_U$ and $X_L$ is greater than or equal to $\gamma$ can be bounded by the following triangular inequality, with the empirical accuracy of one weak labeler in the middle.

$$\Pr[||a_{U,g} - a_{L,g}||_\infty \geq \gamma] \leq \Pr[||a_{U,g} - a_{L,w}||_\infty + ||a_{L,w} - a_{L,g}||_\infty \geq \gamma]$$
$$\leq \Pr[||a_{U,g} - a_{L,w}||_\infty + \epsilon \geq \gamma] \tag{11}$$

Then, since we have $h$ weak labelers, the following union bound holds for an individual labeler $j$ in the committed set and its scalar accuracy.

$$\Pr[||a_{U,g} - a_{L,w}||_\infty + \epsilon \geq \gamma] \leq \sum_{j=1}^{h} \Pr[|a_{U,g,j} - a_{L,w,j}| + \epsilon \geq \gamma] \tag{12}$$

In $X_L$, suppose there are $d_j$ data points not labeled as $abstain$ by labeler $j$, the definition of the empirical accuracy of weak labeler $j$ is

$$a_{L,w,j} = \frac{1}{d_j} \sum_k \mathbf{1}(y_k = \hat{y}_{j,k}) \tag{13}$$

where $y_k$ is the ground truth label of data point $k$ in the non-abstain subset and $\hat{y_{j,k}}$ is the label given by heuristic $j$. By combining Equations 12 and 13, and using Hoeffding's inequality on the independent data points, we can get

$$\Pr[||a_{U,g} - a_{L,w}||_\infty \epsilon \geq \gamma] = \Pr[||a_{U,g} - a_{L,w}||_\infty + \geq \gamma - \epsilon]$$
$$\leq \sum_{j=1}^{h} \Pr[|a_{U,g,j} - a_{L,w,j}| \geq \gamma - \epsilon]$$
$$= \sum_{j=1}^{h} \Pr\left[|a_{U,g,j} - \frac{1}{d_j}\sum_k \mathbf{1}(y_k = \hat{y}_{j,k}|) \geq \gamma - \epsilon\right] \tag{14}$$
$$\leq \sum_{j=1}^{h} 2\exp(-2(\gamma - \epsilon)^2 d_j)$$
$$\leq 2h\exp(-2(\gamma - \epsilon)^2 \min(d_1, \dots, d_h))$$

Equation 14 bounds the probability of Snuba's failure to keep $||a_{U,g} - a_{L,g}||_\infty < \gamma$ in one iteration. Assume without loss of generality that we commit one more weak labeler per iteration, so for the all $h$ iterations, we apply union bound again:

$$\Pr[||a_{U,g} - a_{L,w}||_\infty + \geq \gamma - \epsilon \text{ for any iteration}]$$
$$\leq \sum_{j=1}^{h} \Pr[||a_{U,g} - a_{L,w}||_\infty + \geq \gamma - \epsilon \text{ for one iteration}] \tag{15}$$
$$\leq 2h^2\exp(-2(\gamma - \epsilon)^2 \min(d_1, \dots, d_h))$$

Therefore, if we denote this probability of failure in any iteration as $\delta$, we have the bound

$$\delta \leq 2h^2\exp(-2(\gamma - \epsilon)^2 \min(d_1, \dots, d_h)) \tag{16}$$

and thus obtain Equation 9 by letting $d = \min(d_1, \dots, d_h)$.

## APPENDIX B   WEAK LABELER MODEL HYPERPARAMETERS IN EXPERIMENTS

The LML task configurations are listed in Table 5 for the convenience of replicating our procedure. Because even after hyperparameter search (Section 4.1), there are still differences in the architectures and training hyperparameters throughout a single LML sequence, we have provided shrunken search spaces in weak labeler architecture and training hyperparameters.

| LML Task Sequence | # Tasks | Data Split of $X_{i,L}, X_{i,U}, X_{i,test}$ | Weak Labeler Architecture | Training Hyperparams |
|---|---|---|---|---|
| Binary MNIST | 45 | 120/11880/2000 | Arch A | $lr \in [1\text{e-}4, 1\text{e-}2]$ $n_{batches} \in [5, 10]$ $n_{epochs} \in [30, 100]$ $bootstrap = 30$ |
| Binary CIFAR-10 | 45 | 400/9600/2000 | Arch B | $lr \in [0.8\text{e-}3, 1\text{e-}3]$ $n_{batches} = 30$ $n_{epochs} \in [30, 60]$ $bootstrap = 350$ |
| 10-way CIFAR-10 | 1 | 2000/30000/10000 | Arch B but final $out = 10$ | $lr = 1\text{e-}2$ $n_{batches} = 30$ $n_{epochs} = 50$ $bootstrap = 750$ |
| 5-way CIFAR-100 | 20 | 500/2000/500 | Arch B but final $out = 5$ | $lr \in [0.8\text{e-}3, 1\text{e-}2]$ $n_{batches} = 20$ $n_{epochs} \in [60, 140]$ $bootstrap = 200$ |
| 5-way MNIST | 2 | 150/29750/5000 | Arch A but final $out = 5$ | $lr \in [0.8\text{e-}3, 1.5\text{e-}3]$ $n_{batches} = 10$ $n_{epochs} \in [180, 220]$ $bootstrap = 50$ |

Table 5: LML Task Configurations in Section 5. Please refer to Figure 2 for weak labeler architectures.

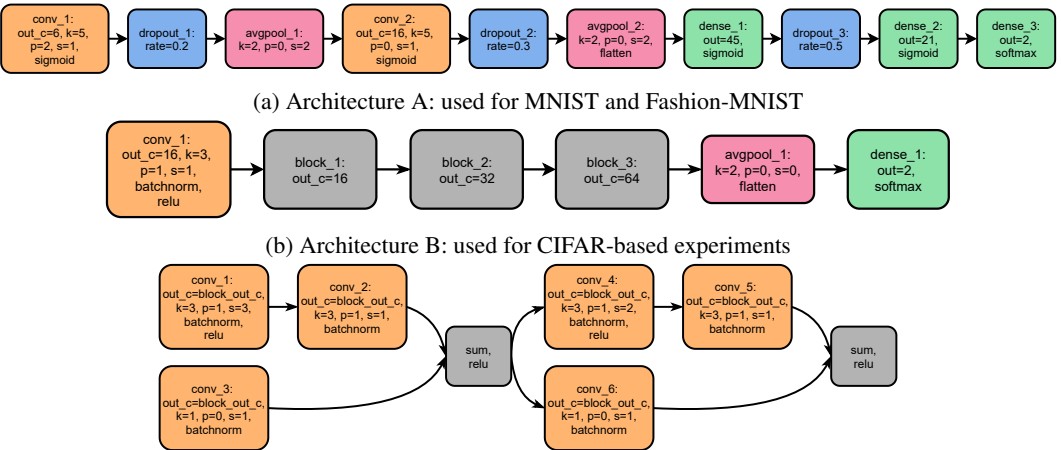

(a) Architecture A: used for MNIST and Fashion-MNIST

(b) Architecture B: used for CIFAR-based experiments

(c) Block structure of Architecture B, where $block\_out\_c$ is the $out\_c$ input into the block

Figure 2: Searched Architectures of Weak Labeling Functions

For example, to re-create our binary MNIST LML tasks, one needs to first arrange the 45 tasks as 0 vs 1, 0 vs 2, ..., 8 vs 9. Then, split the labeled training data, unlabeled training data and testing

data into 120/11880/2000. All the data splits have balanced classes. Finally, perform our procedure described in Section 4, starting from the hyperparameter search in the shrunken search space.

Figure 2 illustrates the searched architectures. The search spaces were inspired by various previous works on CNN designs (Lecun et al., 1998; He et al., 2016). Nonetheless, the final architectures are much smaller since we need fast training of multiple weak labelers. The notations of the figures are: $out\_c$: output channels/number of filters, $k$: size (width and height) of a filter, $p$: padding and $s$: stride.

Figure 3 shows the labeling accuracies of the Mako-labeled data input into LML.

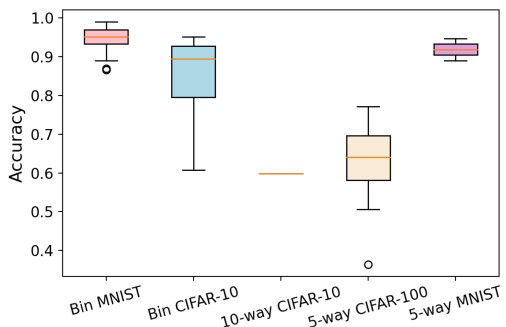

Figure 3: Labeling accuracies of the task sequences after confidence calibration and thresholding, with threshold = $1/num\_classes + 0.01$ for all tasks, i.e. labeling accuracies of $(X_{i,U,c}, Y_{i,U,c}^{\pi})$ described in 4.3

## APPENDIX C    ADDITIONAL SUPERVISED LML EXPERIMENT ANALYSIS

We visualize peak per-task accuracy and final accuracy of supervised LML experiments (Figure 4) and learning curve of final accuracies in these experiments (Figure 5), summarized in Table 1 and 2. In these plots, we used the same notation **L** and **U** of the main text to specify the training data setting, except **TrueU** denoting the case trained on the same instances of **U** but using the true labels instead.

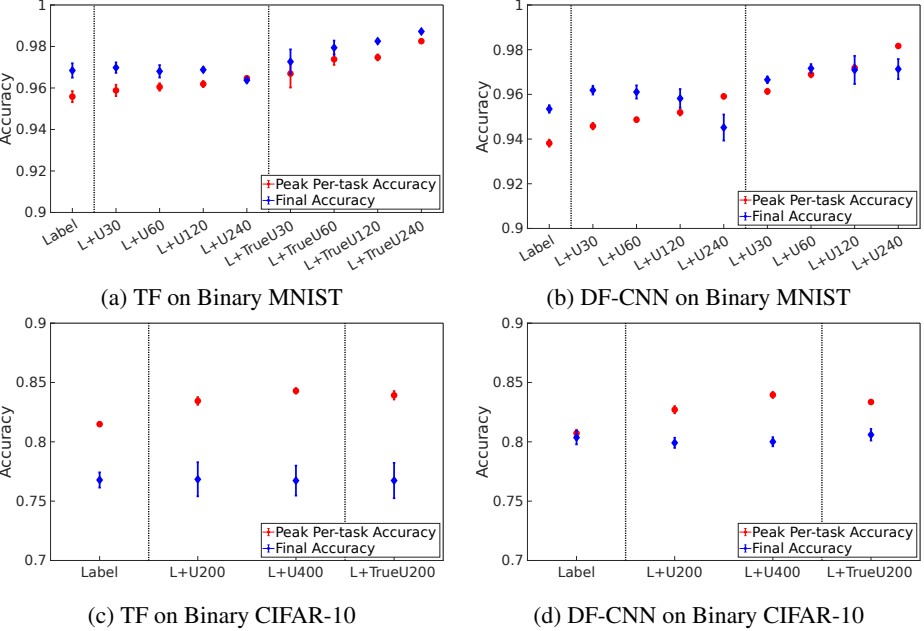

Figure 4: Supervised LML experiments, showing mean and 95% confidence interval. Peak per-task accuracy increases as more mako-labeled data is provided as training, showing the benefit of the generated labels.

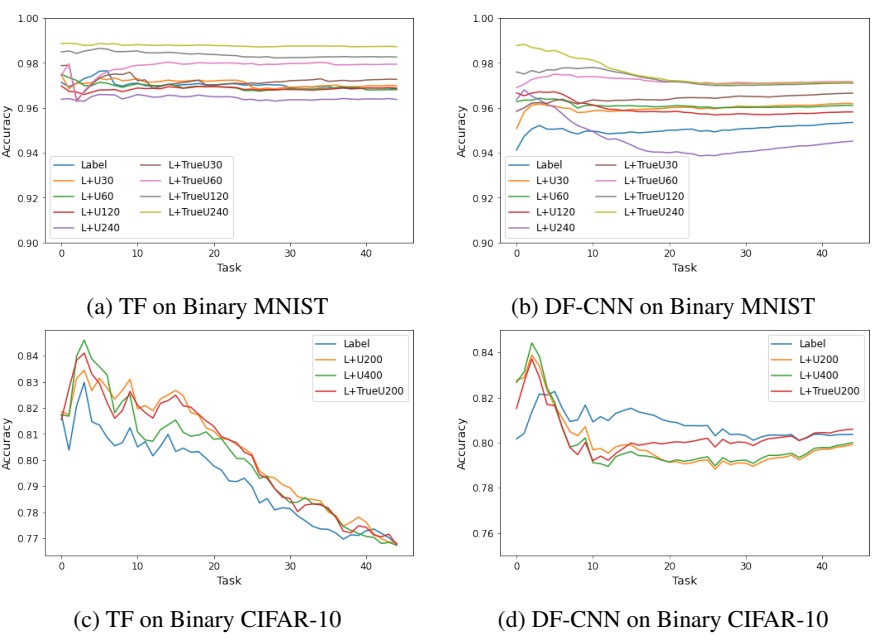

Figure 5: Learning Curve Comparisons

## APPENDIX D  ADDITIONAL SEMI-SUPERVISED LML EXPERIMENT ANALYSIS

We include learning curves for each of the instance-incremental semi-supervised LML experiments in Figure 6 and each of the class-incremental semi-supervised LML experiments in Figure 7. All training curves show final accuracy up to task $i$ as defined in Equation 3, where task $i$ is the current task being trained. For easier tasks with high labeling accuracy, such as CIFAR-10, the Mako-enabled semi-supervised LML methods perform similarly to the equivalent supervised fully-labeled task sequence. Otherwise, the Mako-enabled semi-supervised methods approach fully supervised performance.

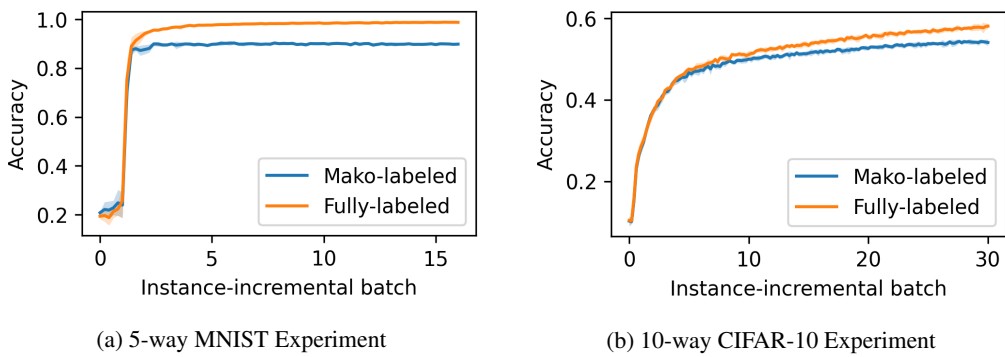

(a) 5-way MNIST Experiment          (b) 10-way CIFAR-10 Experiment

Figure 6: Instance-incremental semi-supervised vs fully supervised learning curve comparisons

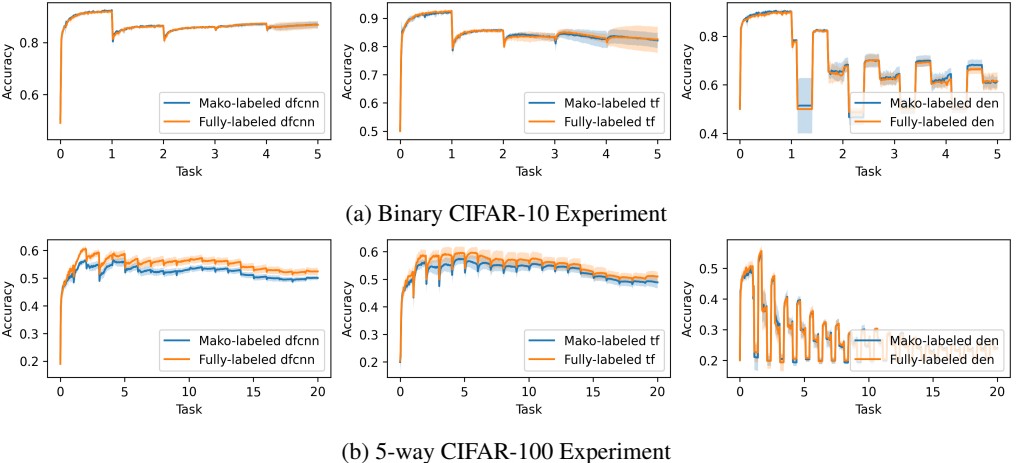

(a) Binary CIFAR-10 Experiment

(b) 5-way CIFAR-100 Experiment

Figure 7: Class-incremental semi-supervised vs fully supervised learning curve comparisons.

We additionally analyze the class-incremental semi-supervised LML baseline experiments for catastrophic forgetting, using the catastrophic forgetting ratio defined in Equation 4. Catastrophic forgetting measures are not available for the baseline methods ORDisCo and DistillMatch as they were not reported in the original publications. The results are shown in Table 6.

For a further breakdown of results, we include task-specific performance measures for all of the semi-supervised LML experiments in Tables 7, 8, 9, and 10.

| | CIFAR-10 | | CIFAR-100 | |
|---|---|---|---|---|
| | Approach | Forget. Ratio | Approach | Forget. Ratio |
| Semi-Supervised | Mako-lab. DF-CNN | **0.98 ± 0.04** | Mako-lab. DF-CNN | **0.85 ± 0.18** |
| | Mako-lab. DEN | 0.72 ± 0.16 | Mako-lab. DEN | 0.48 ± 0.16 |
| | Mako-lab. TF | 0.92 ± 0.09 | Mako-lab. TF | 0.83 ± 0.16 |
| Supervised | *Fully-lab. DF-CNN* | *0.98 ± 0.04* | *Fully-lab. DF-CNN* | *0.83 ± 0.19* |
| | *Fully-lab. DEN* | *0.72 ± 0.17* | *Fully-lab. DEN* | *0.48 ± 0.16* |
| | *Fully-lab. TF* | *0.93 ± 0.10* | *Fully-lab. TF* | *0.82 ± 0.16* |

Table 6: Catastrophic forgetting ratios (see Equation 4) for class-incremental semi-supervised LML experiments, showing mean ± standard deviation

| Approach | Task | Mako-labeled (Semi-supervised) | | Fully-labeled (Supervised) | |
|---|---|---|---|---|---|
| | | Final Acc. | Forget. Ratio | Final Acc. | Forget. Ratio |
| DF-CNN | 0 | 88.1 ± 1.4 | 0.953 ± 0.015 | 86.7 ± 5.3 | 0.944 ± 0.061 |
| | 1 | 75.9 ± 4.8 | 0.942 ± 0.063 | 76.6 ± 1.7 | 0.944 ± 0.024 |
| | 2 | 84.4 ± 2.0 | 0.987 ± 0.020 | 84.7 ± 1.1 | 0.989 ± 0.014 |
| | 3 | 93.2 ± 0.5 | 0.999 ± 0.007 | 93.5 ± 0.7 | 0.999 ± 0.003 |
| | 4 | 92.5 ± 1.1 | 1.000 ± 0.000 | 92.6 ± 0.6 | 1.000 ± 0.000 |
| TF | 0 | 80.5 ± 7.2 | 0.874 ± 0.078 | 79.0 ± 11.2 | 0.853 ± 0.119 |
| | 1 | 67.8 ± 6.7 | 0.845 ± 0.085 | 71.0 ± 5.0 | 0.884 ± 0.064 |
| | 2 | 78.1 ± 8.8 | 0.922 ± 0.105 | 77.4 ± 9.4 | 0.905 ± 0.111 |
| | 3 | 91.3 ± 3.5 | 0.972 ± 0.035 | 92.4 ± 1.4 | 0.987 ± 0.009 |
| | 4 | 92.9 ± 0.5 | 1.000 ± 0.000 | 92.8 ± 1.0 | 1.000 ± 0.000 |
| DEN | 0 | 50.4 ± 0.7 | 0.559 ± 0.013 | 51.7 ± 4.3 | 0.574 ± 0.045 |
| | 1 | 50.2 ± 3.3 | 0.664 ± 0.049 | 50.8 ± 9.4 | 0.678 ± 0.118 |
| | 2 | 58.1 ± 9.7 | 0.716 ± 0.111 | 54.5 ± 8.7 | 0.674 ± 0.106 |
| | 3 | 60.7 ± 7.5 | 0.678 ± 0.086 | 59.9 ± 5.4 | 0.662 ± 0.063 |
| | 4 | 87.4 ± 3.9 | 1.000 ± 0.000 | 90.0 ± 0.6 | 1.000 ± 0.000 |

Table 7: Class-incremental Binary CIFAR-10 experiment results broken down by task, showing mean ± standard deviation

| Approach | Task | Mako-labeled (Semi-supervised) | | Fully-labeled (Supervised) | |
|---|---|---|---|---|---|
| | | Final Acc. | Forget. Ratio | Final Acc. | Forget. Ratio |
| DF-CNN | 0 | 33.9 ± 5.4 | 0.663 ± 0.104 | 31.8 ± 6.0 | 0.583 ± 0.110 |
| | 1 | 35.1 ± 6.5 | 0.547 ± 0.103 | 33.8 ± 6.3 | 0.491 ± 0.093 |
| | 2 | 32.8 ± 4.0 | 0.556 ± 0.067 | 32.8 ± 5.7 | 0.509 ± 0.081 |
| | 3 | 41.6 ± 6.6 | 0.640 ± 0.101 | 38.8 ± 5.9 | 0.562 ± 0.084 |
| | 4 | 37.5 ± 7.5 | 0.530 ± 0.104 | 42.0 ± 7.2 | 0.564 ± 0.092 |
| | 5 | 45.8 ± 5.4 | 0.776 ± 0.080 | 50.7 ± 7.6 | 0.788 ± 0.117 |
| | 6 | 49.3 ± 5.1 | 0.831 ± 0.083 | 51.5 ± 4.5 | 0.804 ± 0.065 |
| | 7 | 46.7 ± 3.5 | 0.768 ± 0.049 | 53.6 ± 3.7 | 0.789 ± 0.066 |
| | 8 | 57.2 ± 3.1 | 0.901 ± 0.048 | 58.2 ± 5.1 | 0.873 ± 0.069 |
| | 9 | 64.2 ± 2.7 | 0.930 ± 0.038 | 65.9 ± 2.3 | 0.923 ± 0.035 |
| | 10 | 70.1 ± 3.1 | 0.944 ± 0.047 | 71.1 ± 3.0 | 0.930 ± 0.040 |
| | 11 | 53.0 ± 3.5 | 0.917 ± 0.055 | 57.7 ± 2.5 | 0.952 ± 0.046 |
| | 12 | 61.7 ± 1.7 | 0.967 ± 0.033 | 64.2 ± 1.7 | 0.963 ± 0.033 |
| | 13 | 54.1 ± 1.8 | 0.970 ± 0.034 | 55.4 ± 1.8 | 0.922 ± 0.029 |
| | 14 | 33.0 ± 1.7 | 0.987 ± 0.032 | 36.7 ± 1.9 | 0.977 ± 0.039 |
| | 15 | 50.0 ± 1.3 | 1.006 ± 0.029 | 52.7 ± 2.5 | 0.971 ± 0.035 |
| | 16 | 48.1 ± 2.2 | 1.002 ± 0.024 | 50.0 ± 2.3 | 0.967 ± 0.021 |
| | 17 | 53.1 ± 1.2 | 1.000 ± 0.013 | 56.1 ± 1.6 | 0.994 ± 0.018 |
| | 18 | 62.7 ± 2.1 | 1.001 ± 0.015 | 67.7 ± 1.8 | 0.996 ± 0.009 |
| | 19 | 70.5 ± 1.8 | 1.000 ± 0.000 | 77.3 ± 1.9 | 1.000 ± 0.000 |

Table 8: Class-incremental 5-way CIFAR-100 experiment, DF-CNN, results broken down by task, showing mean ± standard deviation

| Approach | Task | Mako-labeled (Semi-supervised) | | Fully-labeled (Supervised) | |
|---|---|---|---|---|---|
| | | Final Acc. | Forget. Ratio | Final Acc. | Forget. Ratio |
| | 0 | $36.9 \pm 6.7$ | $0.729 \pm 0.135$ | $38.8 \pm 9.1$ | $0.736 \pm 0.174$ |
| | 1 | $43.4 \pm 4.8$ | $0.692 \pm 0.082$ | $49.9 \pm 9.9$ | $0.750 \pm 0.138$ |
| | 2 | $36.2 \pm 7.0$ | $0.639 \pm 0.122$ | $37.5 \pm 7.0$ | $0.617 \pm 0.116$ |
| | 3 | $56.4 \pm 5.2$ | $0.870 \pm 0.085$ | $54.0 \pm 5.5$ | $0.791 \pm 0.072$ |
| | 4 | $40.6 \pm 13.2$ | $0.585 \pm 0.193$ | $42.4 \pm 10.7$ | $0.600 \pm 0.151$ |
| | 5 | $51.0 \pm 7.4$ | $0.839 \pm 0.124$ | $50.9 \pm 13.6$ | $0.760 \pm 0.206$ |
| | 6 | $46.3 \pm 7.1$ | $0.767 \pm 0.126$ | $52.6 \pm 5.0$ | $0.832 \pm 0.085$ |
| | 7 | $48.8 \pm 9.3$ | $0.804 \pm 0.152$ | $50.8 \pm 7.4$ | $0.804 \pm 0.081$ |
| | 8 | $49.6 \pm 8.7$ | $0.773 \pm 0.111$ | $47.0 \pm 8.5$ | $0.732 \pm 0.122$ |
| | 9 | $60.4 \pm 7.1$ | $0.885 \pm 0.106$ | $58.5 \pm 6.0$ | $0.828 \pm 0.095$ |
| TF | 10 | $57.5 \pm 11.4$ | $0.777 \pm 0.154$ | $56.9 \pm 14.0$ | $0.732 \pm 0.180$ |
| | 11 | $48.3 \pm 8.3$ | $0.842 \pm 0.140$ | $46.9 \pm 6.0$ | $0.798 \pm 0.115$ |
| | 12 | $53.6 \pm 5.2$ | $0.881 \pm 0.072$ | $57.6 \pm 3.8$ | $0.905 \pm 0.064$ |
| | 13 | $48.4 \pm 5.5$ | $0.888 \pm 0.097$ | $52.9 \pm 8.2$ | $0.847 \pm 0.126$ |
| | 14 | $31.0 \pm 2.0$ | $0.924 \pm 0.070$ | $31.6 \pm 2.9$ | $0.945 \pm 0.090$ |
| | 15 | $41.5 \pm 6.0$ | $0.836 \pm 0.134$ | $48.1 \pm 2.8$ | $0.928 \pm 0.046$ |
| | 16 | $45.1 \pm 3.0$ | $0.956 \pm 0.073$ | $45.2 \pm 5.2$ | $0.936 \pm 0.093$ |
| | 17 | $50.0 \pm 4.3$ | $0.939 \pm 0.083$ | $54.5 \pm 2.4$ | $0.969 \pm 0.033$ |
| | 18 | $62.8 \pm 2.7$ | $0.974 \pm 0.032$ | $67.8 \pm 2.2$ | $0.962 \pm 0.023$ |
| | 19 | $69.6 \pm 2.2$ | $1.000 \pm 0.000$ | $76.4 \pm 2.7$ | $1.000 \pm 0.000$ |

Table 9: Class-incremental 5-way CIFAR-100 experiment, TF, results broken down by task, showing mean $\pm$ standard deviation

| Approach | Task | Mako-labeled (Semi-supervised) | | Fully-labeled (Supervised) | |
|---|---|---|---|---|---|
| | | Final Acc. | Forget. Ratio | Final Acc. | Forget. Ratio |
| | 0 | $21.1 \pm 2.9$ | $0.433 \pm 0.059$ | $19.6 \pm 1.1$ | $0.388 \pm 0.033$ |
| | 1 | $18.6 \pm 2.4$ | $0.373 \pm 0.061$ | $22.7 \pm 2.3$ | $0.455 \pm 0.092$ |
| | 2 | $20.3 \pm 2.2$ | $0.424 \pm 0.056$ | $21.0 \pm 2.4$ | $0.477 \pm 0.097$ |
| | 3 | $20.1 \pm 1.6$ | $0.381 \pm 0.080$ | $20.5 \pm 3.0$ | $0.361 \pm 0.058$ |
| | 4 | $24.5 \pm 4.7$ | $0.420 \pm 0.106$ | $20.1 \pm 1.5$ | $0.373 \pm 0.069$ |
| | 5 | $21.9 \pm 1.4$ | $0.444 \pm 0.079$ | $21.8 \pm 2.5$ | $0.423 \pm 0.047$ |
| | 6 | $19.3 \pm 1.0$ | $0.405 \pm 0.110$ | $19.6 \pm 2.7$ | $0.350 \pm 0.062$ |
| | 7 | $22.4 \pm 2.5$ | $0.490 \pm 0.171$ | $21.8 \pm 2.3$ | $0.452 \pm 0.095$ |
| | 8 | $22.0 \pm 1.3$ | $0.451 \pm 0.058$ | $21.8 \pm 1.3$ | $0.472 \pm 0.058$ |
| | 9 | $24.0 \pm 2.6$ | $0.421 \pm 0.077$ | $22.0 \pm 2.7$ | $0.402 \pm 0.062$ |
| DEN | 10 | $22.7 \pm 1.8$ | $0.350 \pm 0.048$ | $23.1 \pm 3.1$ | $0.359 \pm 0.046$ |
| | 11 | $22.9 \pm 1.9$ | $0.502 \pm 0.058$ | $22.8 \pm 2.5$ | $0.477 \pm 0.050$ |
| | 12 | $21.9 \pm 2.1$ | $0.408 \pm 0.073$ | $22.3 \pm 1.4$ | $0.432 \pm 0.041$ |
| | 13 | $24.4 \pm 3.0$ | $0.506 \pm 0.056$ | $24.4 \pm 3.3$ | $0.494 \pm 0.058$ |
| | 14 | $20.9 \pm 1.5$ | $0.689 \pm 0.082$ | $20.2 \pm 1.1$ | $0.737 \pm 0.083$ |
| | 15 | $22.8 \pm 3.0$ | $0.497 \pm 0.068$ | $20.9 \pm 1.6$ | $0.479 \pm 0.051$ |
| | 16 | $21.2 \pm 0.8$ | $0.516 \pm 0.059$ | $21.1 \pm 2.1$ | $0.527 \pm 0.065$ |
| | 17 | $22.2 \pm 1.6$ | $0.429 \pm 0.033$ | $21.6 \pm 1.7$ | $0.434 \pm 0.038$ |
| | 18 | $23.1 \pm 2.1$ | $0.416 \pm 0.065$ | $23.0 \pm 2.0$ | $0.446 \pm 0.064$ |
| | 19 | $65.2 \pm 3.5$ | $1.000 \pm 0.000$ | $64.1 \pm 5.2$ | $1.000 \pm 0.000$ |

Table 10: Class-incremental 5-way CIFAR-100 experiment, DEN, results broken down by task, showing mean $\pm$ standard deviation

