# OpenReview forum: "Mako: Semi-supervised continual learning with minimal labeled data via data programming"
_ICLR.cc/2022/Conference — ICLR 2022 Submitted_

### Official Review · Reviewer_Fba4 · 2021-10-22

**Correctness:** 3
**Technical Novelty And Significance:** 1
**Empirical Novelty And Significance:** 2
**Recommendation:** 3
**Confidence:** 4

**Main Review:**

Strengths:

*  The paper is clearly written and the topic is relavent.
*  Experiments on  several different scenarios


Weakness:
* The technical novelty of proposed methods is very limited. The method is simply the combination of existing methods. For example, the classifer is a collection of weak labeling functions with bootstrapping, e.g., Snuba. In addition, the data programming techniques are not novel as well. It is very similar to the self-training [1], which predicts the labels of unlabeled data during the training process.  The self-training techniques are widely adopted in semi-supervised learning. It would be better to discuss in related work for the difference between the proposed method and self-training and compare to state of art self-training methods in experiment.

* There is lack of insights on the proposed method for mitigate catastrophic forgetting, which is main focus of this paper.

* For the evaluation metrics, the author proposed  catastrophic forgetting ratio up to task i for evaluating the forgetting. However, for most continual learning works, they commonly adopt backward transfer [2] to evaluate the forgetting issue. It would be bettter to show the commonly adopted metrics for a direct comprions to related works.

* For the experiment,  improvement on forgetting seems quite marginal, although the peak per-task accuracy is improved. Also, seems that there are many hyperparameters in the proposed method. It would be great to show the sensitivity of the hyperparameters.


Reference:
[1] Uncertainty-aware Self-training for Text Classification with Few Labels. Subhabrata Mukherjee and Ahmed Hassan Awadallah. NeurIPS 2020
[2] On Tiny Episodic Memories in Continual Learning, Arslan Chaudhry, et al. https://arxiv.org/abs/1902.10486

**Summary Of The Paper:**

This paper proposes  data programming method, named Mako,  for semi-supervised continual learning. Mako automatically generates labels for unlabeled data with a set of weak labeling functions, each of the functions is trained on subset of training set with bootstrapping. Experimental on several datasets demonstrate the effectiveness of the proposed methods.

**Summary Of The Review:**

The paper is clearly written, but the technical novelty is quite limited. The analysis and experiments also need to be significantly improved.

---

> ### Author Response · Authors · 2021-11-18
> **Response to reviewer Fba4**
>
> We thank the reviewer for the thorough and detailed feedback, and we appreciate their acknowledgement of our efforts on problem identification, paper writing and experiments. To clarify the concerns:
>
> (1) The technical novelty of the proposed method is limited.
> Our original work is the Mako framework described in Section 4. It is not simply using previous work, but a modification of each individual component in order to assemble them to fit the nature of LML tasks. That is, previous data programming tools are useful for isolated ML, but since LML has additional requirements to learn over task sequences, we extend data programming methods by using deep learning-based weak labeler architectures, bootstrapping, adding automatic hyperparameter search and calibration, all of which are not used in previous data programming works. Our goal is to build a semi-supervised LML framework with high performance as well as high modularity, which means components are not intertwined and can be individually upgraded/replaced.
>
> (2) Lack of insights on the proposed method for mitigating catastrophic forgetting.
> Mitigating catastrophic forgetting is an important concern for continual learning, and as such we include it as one of our performance measures.  Our results show that leveraging unlabeled data with Mako does not significantly impact the catastrophic forgetting ratio when compared to either a small labeled data set (Tables 1 and 2) or a full-size labeled data set (Table 6).  Based on these results we conclude that improvements to catastrophic forgetting come mainly from the lifelong learning method, not the Mako labeling process.  Since, as we show in the paper, Mako is agnostic to lifelong learning methods, the fact that using it does not impact catastrophic forgetting mitigation allows it to be used with lifelong learning methods without adding additional concerns about catastrophic forgetting.  We will make these insights clear in our discussion of the results.
>
> (3) Use other metrics, such as backward transfer, for evaluation.
> We are working on more comprehensive evaluation on more metrics such as precision and labeling function coverages. Catastrophic forgetting ratio, used in some baselines of semi-supervised lifelong learning, indicates backward transfer normalized by the performance on the task after learned, although we do appreciate the suggestion and can update our experiments to directly compute additional lifelong learning metrics.
>
> (4) Improvement on forgetting seems quite marginal and concerns on hyperparameters.
> Although our framework shows clear improvements in final accuracy, both in relation to the labeled data set only (“L” in Table 1) and when compared to ORDisCo and DistillMatch in Section 5.3, we see similar catastrophic forgetting performance with respect to the small labeled data set (Tables 1 and 2) or with respect to a full-size labeled data set (Table 6). Taken together, our metrics and experiments show that leveraging unlabeled data with Mako improves accuracy without impacting catastrophic forgetting. We plan to analyze hyperparameter sensitivity and add those results to an appendix before final submission of this paper.

---

> > ### Comment · Reviewer_Fba4 · 2021-11-29
> > **rebuttal update**
> >
> > Thank you for your update. I read the rebuttal and other reviewers' comments and still think that my concerns have not been sufficiently addressed, so I will keep my previous score.

---

### Official Review · Reviewer_ECpE · 2021-11-01

**Correctness:** 4
**Technical Novelty And Significance:** 3
**Empirical Novelty And Significance:** 3
**Recommendation:** 5
**Confidence:** 5

**Main Review:**

#### Pros
Continual learning with fewer annotated data is an important research task. The proposed framework uses data programming techniques to generate pseudo labels for unlabeled data and makes the task suitable for various fully supervised LML methods. Compared with fully supervised methods, the experimental results seem promising.

#### Cons
1. Since most benefits come from the data programing techniques (Snuba in this paper), I would like to see more analysis on the annotation process, performance of down-stream tasks is only one of the metrics, e.g.,
	a. It is unclear that how many labeling functions are generated in each task? Precision and coverage of each LF in the committed set are more meaningful than labeling accuracy in Fig.3.
	b. One key fact of labeling functions in Snuba is diversity. Although the authors trained labeling functions on different sampled sets, the laziness of the neural network makes it prefer to output similar results. How different are the results obtained from different labeling functions? It would be nice to see a numeric comparison.
	c. It seems a sophisticated process to annotate unlabeled data. It would be better to provide/plot average iterations and computational times for [hyperparameter search <--> data programming] for each task.

2. I’m concerned about the module in Sec 4.1 and Sec 4.2:
	a. I am not sure whether I grasped the main points of the automatic hyperparameter search and data programming. Does the automatic hyperparameter search module first search the best hyperparameter as a fixed heuristic pattern, then the DP module generates different LF with the same hyperparameters but different sampled subsets?
	b. How many heuristics are generated for pruning.
c. Some important technical details are not clear in this paper, e.g., the hyperparameters in Snuba.

#### Misc
1. Just for curiosity, why do both Mako-labeled DEN and fully-labeled DEN achieve worse performance than the baselines?
2. I have a doubt about the experimental results of DEN in the last row of Table 7 and Table 10: What’s the reason for the performance booming in the last task?

#### Minor points
“purposed” in the 1st line of the second paragraph in Sec.2 should be “proposed”?


**Summary Of The Paper:**

This paper presents an interesting idea of using data programming techniques to enable continual semi-supervised learning with limited labeled data. It proposes a stage-wise pipeline where probabilistic pseudo labels are first produced by a Snuba based Data Programming framework, then calibrates them by the temperature scaling, and finally inputs into the mounted Lifelong Machine Learning (LML) tools. Experiments show that the proposed framework achieves similar performance to fully supervised methods.

**Summary Of The Review:**

In conclusion, despite the successful improvement of the LML task, I still have lots of concerns about the labeling results, which is the key to improvements. I would be more convinced if more analysis of labeling results could be done in this paper. For the current version, I recommend a negative score as my initial rating.

---

> ### Author Response · Authors · 2021-11-18
> **Response to reviewer ECpE**
>
> We thank the reviewer for the thorough and thoughtful comments, and we very appreciate the acknowledgement of our effort on problem identification and experiments. To answer the concerns raised:
>
> (1) Would like to see more analysis on the annotation process.
> We are currently working on more comprehensive evaluations of Mako on more metrics, such as precision and coverage. We appreciate the point that diversity of labeling functions is also an important metric, and we can measure it by, for example, average L-p norm distance among the label vectors generated. Computational overhead analysis is also considered for the next step. We will add the data programming hyperparameters, such as these for modified Snuba, if we have a chance to edit this paper before the final results.
>
> (2)  Main points of hyperparameter search and data programming.
> The hyperparameter search follows a fixed heuristics as described in 4.1. It searches for the optimal hyperparameters for 10 randomly bootstrapped subsets of data, so that these hyperparameters are considered optimal for all subsets in general and we use them as default in data programming. However, as explained in 4.2, if the theoretical exit condition is triggered before sufficient weak labelers are generated, we will search again on the remaining hyperparameter search space and repeat with the new optimum.
>
> (Misc.) The final accuracy measured per task in Tables 7 and 10 are computed at the end of training the entire task sequence.  DEN exhibits a high degree of catastrophic forgetting, i.e. training additional tasks interferes with the performance of previously learned tasks, and so the final task in the sequence has a large jump in performance since there are no tasks trained subsequently, and thus no interference.  We’re also curious why DEN performs noticeably worse than all of the other methods.  We reached out to the original authors of DEN and are running their code after hyperparameter tuning, and these are the best results we could achieve with their method.

---

### Official Review · Reviewer_E934 · 2021-11-02

**Correctness:** 3
**Technical Novelty And Significance:** 2
**Empirical Novelty And Significance:** 3
**Recommendation:** 5
**Confidence:** 3

**Main Review:**

Strength:
The definition of the problem, the applicable scenarios and the description of the methods are very clear in this paper. The scenario that the author want to solve where labeled training data is expensive to obtain in some lifelong machine learning tasks is also interesting.   Under the premise of this scenario, the author built a processing framework, completing a series of functions such as hyper-parameters search, pseudo labeling and so on. Experimental results also demonstrate its effectiveness.

Weakness:

1）	I am confused about the definition of some methods in the article. For example, What does the “weak labeler” refer to in the article?  Does the definition of this part correspond to “weak augmentation” in the semi-supervised problem?. I don’t know what is the “weak label” in classification problem. I hope the author can give some description about this.

2）	The article spends a lot of time on the definition of various issues and contexts, It's not even possible for me to figure out where the work is the original work of this article. Even in Sec.4, I found that it seemed to be integrating some of the previous working methods to form a system framework, such as data programming, confidence calibration and so on. It makes me feel that this article is more like a technical report rather than a research work. May be I’m misunderstanding about it, So I hope the author can clearly state the contribution of this work.

3）	The paper proposed a framework based on semi-supervised continual learning, the main statement of the article also lies in the application of semi-supervised learning. However, some description about semi-supervised is hard to understand. For example, the most important problem in semi-supervised learning is how to obtain accurate pseudo labels, but I don’t see how the author solves this problem, only by using a previous method data programming? Or does the article just introduce the semi supervised learning into the some scenarios of continuous learning?  If it is the latter, there are many more mature semi supervised algorithms, can you explain data programming is used?


**Summary Of The Paper:**

This paper propose a wrapper tool that mounts on top of supervised Lifelong Machine Learning (LML) frameworks, leveraging a well-known method data programming. The contributions of this paper can be summarized in three aspects. 1)  Adapting automatic label generation by semi-supervised learning/data programming to LML in some special scenario. 2) Implementing a LML wrapper that can accomplish some tasks under some restrictions. 3) Through detailed experimental results prove the superiority of its method.

**Summary Of The Review:**

This paper implemented a semi-supervised continual learning framework that can be mounted on top of any existing LML tool. It integrates some methods to solve the scenario where labeled training data is expensive. The problems and application scenarios proposed in this paper are of practical significance. However, in my opinion, the engineering significance of the paper is greater than its academic significance, it lacks of novelty in the view of research.

---

> ### Author Response · Authors · 2021-11-18
> **Response to reviewer E934**
>
> We thank the reviewer for their detailed and thoughtful feedback. We appreciate the acknowledgement of our strengths in problem formulation, framework designing and experiments. To answer their concerns:
>
> (1) Confusions about some terms in this article such as “weak labeler”.
> “Weak labeler” is a term used in data programming/weak supervision, which describes a machine learning model that performs only slightly better than random guessing, but such labelers can become powerful when many are ensembled. This is the approach used by data programming to automatically generate pseudo labels. Please refer to our citation “A. Ratner et al. 2016b” (Data programming: Creating large training sets, quickly) for more definitions of relevant terms.
>
> (2) Confusions about what is the original work.
> Our original work is the Mako framework described in Section 4. It is not simply using previous work, but a modification of each individual component in order to assemble them to fit the nature of LML tasks. That is, previous data programming tools are useful for isolated ML, but since LML has additional requirements to learn over task sequences, we extend data programming methods by using deep learning-based weak labeler architectures, bootstrapping, adding automatic hyperparameter search and calibration, all of which are not used in previous data programming works. Our goal is to build a semi-supervised LML framework with high performance as well as high modularity, which means components are not intertwined and can be individually upgraded/replaced.
>
> (3) Some description about semi-supervised is hard to understand.
> As answered in (2), our solution to this problem is the Mako framework, where the core solution component is data programming, which is significantly extended from previous work. While there are other approaches to semi-supervised machine learning, there are very few approaches suitable for semi-supervised continual or lifelong machine learning (LML).  For existing semi-supervised LML approaches (CNNL, ORDisCo, and DistillMatch), we show comparisons experiments (Section 5.3), and we show that Mako outperforms them with assistance of unlabeled data in both instance- and class-incremental scenarios.  We are happy to include additional semi-supervised lifelong machine learning methods if the reviewers have specific suggestions.

---

> ### Comment · Reviewer_E934 · 2021-11-30
> **Response to authors' feedback**
>
> Thanks for the authors’ responses, after reading the rebuttal and other reviewers’ comments, I still hold my initial ratings.

---

### Official Review · Reviewer_XNuq · 2021-11-02

**Correctness:** 2
**Technical Novelty And Significance:** 2
**Empirical Novelty And Significance:** 2
**Recommendation:** 3
**Confidence:** 4

**Main Review:**

Pros:

[1] The lifelong learning without catastrophic forgetting is a critical and interesting issue, the method proposed in this paper is shown to be effective in addressing this issue on some small-scale benchmarks, such as MINIST, CIFAR-10 and CIFAR-100.

[2] The authors conducted comprehensive experiments on LML and proposed several evaluation metrics to quantitatively measure the ability of MAKO in eliminating catastrophic forgetting, improving peak per-task accuracy and final accuracy.

[3] Exploring the lifelong learning using a limited amount of data, i.e. semi-supervised LML, is a new setting and of great interest.

[4] The paper is well-organized.

Cons:

[1] Although authors conducted comprehensive experiments on MNIST and CIFAR, the experiments on larger benchmarks may be necessary to show the improvements obtained through using MAKO.

[2] Most of the components of MAKO are based on the prior arts proposed in previous literatures, for example, the method for labeling training data with weak supervision is based on Snuba, therefore, the technical contribution of this paper may be limited.

[3] If the method is resistance to catastrophic forgetting, why the forgetting ratio is always lower than baseline method? Does it mean that there is less positive knowledge transfer from the later tasks to the earlier ones?

[4] In table 1, the more unlabeled data is used during the training process, the worse the final accuracy is, which is quite confusing. The final accuracy and forget ratio are both lower than the baseline method when 40 unlabeled instances are used. If the proposed method is truly effective and is able to avoid catastrophic forgetting, why the performance is lower?

This metric is less than 1 if the LML model loses its performance on the earlier tasks, and it is greater than 1 if there is positive knowledge transfer from the later tasks to the earlier ones.


**Summary Of The Paper:**

In contrast to previous works on lifelong machine learning (LML) that put their focus on the supervised learning settings, this paper concentrates on the scenario that only a limited amount of data is available. The proposed method MAKO is mounted on the top of supervised LML model, without introducing additional knowledge based overhead, for better leveraging the unlabeled data. Labeling new data can be realized by using the data programming method which is supervised by the labeled data. The target of this paper is to design a SSL LML framework that minimizes the performance between using partially labeled data, and the upper-bound performance using fully labeled data. Several experiments on standard image classification data sets including MNIST, CIFAR-10 and CIFAR-100 are used to evaluate the the effectiveness of MAKO.

**Summary Of The Review:**

I like the idea of semi-supervised lifelong machine learning and I agree with the authors that previous works on LML only consider the fully-supervised lifelong learning, which require the use of all labeled data and is quite expensive. However, the technical contribution of this paper may be limited. Considering that the performance of the proposed method is not that satisfying either, my current rate of this paper is: "3: reject, not good enough".

---

> ### Author Response · Authors · 2021-11-18
> **Response to reviewer XNuq**
>
> We thank the reviewer for the thoughtful comments, and we appreciate their acknowledgement of our effort, specifically on exploring LML with limited data and labels, conducting experiments and organizing the paper. To answer their concerns:
>
> (1) The framework needs experiments on larger benchmarks.
> We used MNIST and CIFAR for experiments due to the direct comparison to previous semi-supervised lifelong learning works, and we showed the improvements of performance of lifelong learners in terms of final accuracy and speed of convergence (Table 3 and 4). We are also experimenting on additional benchmarks including Omniglot, which consist of larger amounts of data and feature spaces.
>
> (2) The technical contribution is limited.
> For this paper, we aim to construct a semi-supervised LML tool with high modularity, which means the components are not intertwined and can be easily replaced. The labeling component is not simply Snuba, but a modified version, with different weak labeler architecture, hyperparameter search and calibration, that can fit the LML tasks due to the tasks’ highly dynamic nature. For future work, because of the modularity, this labeler component can be further augmented by optimal transport distance or other features as discussed at the end of our paper. The goal is to complete high performance LML sequences with low data and labeling cost.
>
> (3) Why is the forgetting ratio always lower than the baseline method? Does it mean that there is less positive knowledge transfer from the later tasks to the earlier ones?
> Yes, forgetting ratios less than 1.0 indicate less positive reverse transfer between tasks, but the differences in our results are marginal.  Any resistance to catastrophic forgetting (shown by forgetting ratios around 1.0) is important in continual learning. Our results show that Mako does not significantly impact the catastrophic forgetting ratio when compared to either a small labeled data set (Tables 1 and 2) or a full-size labeled data set (Table 6). These results must be taken together with other metrics like final accuracy, showing that Mako leverages unlabeled data to improve final accuracy while remaining resistant to catastrophic forgetting.
>
> [4] In table 1, the more unlabeled data is used during the training process, the worse the final accuracy is.
> The goal of our tool is to show that with additional unlabeled data, we can achieve better results than using the labeled data only, and we show that the performance is better than the baseline (“L” in Table 1) for smaller amounts of unlabeled data. However, since the labeling component can not achieve 100% accuracy, and the unlabeled data set is usually significantly larger than the labeled, adding large amounts of unlabeled data can cause performance degradation. This raises an interesting future question which we are exploring: what is the optimal ratio of labeled vs. unlabeled data given our tool.

---

### Decision · Program_Chairs · 2022-01-20

**Decision:**

Reject

**Comment:**

This submission proposes "Mako", which enables continual learning when only a limited amount of labeled data is available (along with a good deal of unlabeled data). Reviewers shared concerns about difficulty in understanding which components of the proposed system were novel, especially given that the most important components seemed to be proposed in past work. Reviewers also had difficulty getting insight on which parts of the system were most useful, and further requested additional experiments on harder benchmarks. There consensus was therefore to reject the paper.